



# Climate change impacts on snow and streamflow drought regimes in four ecoregions of British Columbia

Jennifer R. Dierauer[1], Diana M. Allen[2], Paul H. Whitfield[2,3,4]

[1]College of Natural Resources, University of Wisconsin-Stevens Point, Stevens Point, WI 54481.
[2]Department of Earth Sciences, Simon Fraser University, Burnaby, V5A 1S6, Canada.
[3]Centre for Hydrology, University of Saskatchewan, Saskatoon, SK.
[4]Environment and Climate Change Canada, Vancouver, BC.

*Correspondence to*: Jennifer R. Dierauer (jbrand@uwsp.edu)

**Abstract.**

In many regions with seasonal snow cover, summer streamflow is primarily sustained by groundwater that is recharged during the snowmelt period. Therefore, below-normal snowpack (snow drought) may lead to below-normal summer streamflow (streamflow drought). Summer streamflow is important for supplying human needs and sustaining ecosystems. Climate change impacts on snow have been widely studied, but the relationship between snow drought and streamflow drought is not well

understood. In this study, a combined investigation of climate change impacts on snow drought and streamflow drought was completed using generic groundwater – surface water models for four headwater catchments in different ecoregions of British Columbia. Results show that, in response to increased precipitation and temperature, the snow drought regime changes substantially for all four catchments. Warm snow droughts, which are caused by above-normal winter temperatures, increase in frequency, and dry snow droughts, which are caused by below-normal winter precipitation, decrease in frequency. The shift

toward more frequent and severe temperature-related snow droughts leads to decreased summer runoff, decreased summer groundwater storage, and more extreme low flows in summer. Moreover, snow droughts propagate into summer streamflow droughts more frequently in the future time periods (2050s, 2080s) as compared to the baseline 1980s period. Thus, warm snow droughts not only become more frequent and severe in the future but also more likely to result in summer streamflow drought conditions.

**1 Introduction**

If temperatures rise as expected (Intergovernmental Panel on Climate Change [IPCC], 2013), precipitation will be more likely to fall as rain than to fall as snow. Warming alone is expected to have large impacts on the hydrologic regimes of catchments with seasonal snow cover (Barnett et al., 2005), with decreased annual snowpack leading to earlier snowmelt and diminished and potentially warmer late summer flows (Barnett et al., 2008; Seager et al., 2013; Godsey et al, 2014; Reynolds

et al., 2015; Service, 2015; Jenicek et al., 2016). Snowmelt is more effective than rainwater at infiltrating beyond the root



zone (Earman et al., 2006), and the spring snowmelt pulse often makes up a large fraction of the groundwater recharge in seasonally snow-covered catchments (Winograd et al., 1998; Earman et al., 2006; Ajami et al., 2012). This sustained pulse of groundwater recharge from snowmelt is an important component of the hydrologic regime, and the spring groundwater recharge plays a key role in sustaining streamflow during summer low flow period. A lack of snow accumulation in winter,

i.e. a "snow drought", thus has important implications for water quantity in the following summer season when water demands are high.

Snow drought (Ludlum, 1978; Wiesnet, 1981) can be caused by below-normal precipitation and/or above-normal temperatures (Harpold et al., 2017), with both types leading to below-normal summer streamflow (Harpold et al., 2017; Dierauer et al., 2018). Streamflow droughts can propagate directly from snow droughts, with warm and dry winters

producing longer, more severe low flow periods in the summer (Dierauer et al., 2018). While the role of temperature in snowmelt hydrology has been widely studied (Leith and Whitfield, 1998; Whitfield and Cannon, 2000; Adam et al., 2009; Déry et al., 2009; Pederson et al., 2011; among others), no studies have explicitly quantified the impact of different snow drought types, i.e. dry, warm, or warm and dry, on the severity of summer streamflow droughts. Moreover, no studies have completed a combined analysis of snow drought and streamflow drought regimes in the context of climate change.

Recent work by Dierauer et al. (2019) has shown that temperature-related snow drought risk increases with increasing mean winter temperatures and identified temperature thresholds above which hydroclimatic change "accelerates". As temperatures rise, increased frequency and severity of temperature-related snow drought will likely lead to increased frequency and severity of summer streamflow droughts. The magnitude of these changes, however, will likely depend on a catchment's starting point, i.e. its baseline mean winter temperature. To investigate climate change impacts on snow drought and the

subsequent impacts on summer streamflow drought, this study combines climate change projections with generic groundwater - surface water models for four headwater catchments located in different ecoregions of British Columbia, Canada. These headwater catchments span a large range of baseline climate conditions and, thus, should exhibit different responses to climate warming. The study locations and the reasons for choosing each catchment are discussed in Sect. 2. The development of groundwater-surface water models, the choice of climate change scenarios, and the assessment of low flows and snow drought

are discussed in Sect. 3. Results are presented in Sect. 4 and discussed in Sect. 5, and conclusions are stated in Sect. 6.

## 2 Study Locations

Four headwater catchments spanning a range of climate conditions were chosen for this study – each is located in a different level I ecoregion (Commission for Environmental Cooperation [CEC], 2011) of British Columbia (Table 1, Figure 1), and each represents a municipal, agricultural, or industrial surface water supply source. The Fort Nelson River headwater

catchment is the coldest and has a mean annual temperature of -0.6°C. The Fort Nelson catchment is located in the Hay and Slave River Lowlands of the Taiga ecoregion. The Blueberry River headwater catchment is the second coldest and has a mean annual temperature of 0°C. The Blueberry catchment is located in the Clear Hills and Western Alberta Uplands of the Northern





Forests ecoregion. Both catchments have relatively cold, dry winters, and, on average, receive less than 15 cm of snow per year. Additionally, both catchments are in Northeast British Columbia (NEBC) – an area of expanding shale gas development where multi-stage hydraulic fracturing operations require large quantities of water. In 2015, 45% of the 7.74 million $m^3$ of water used for hydraulic fracturing in NEBC was sourced from surface water (British Columbia Oil and Gas Commission, 5 2016), with the highest water demand occurring in the warmer months from May to September.

Whiteman Creek headwater catchment has a mean annual temperature of 2.1°C and is located in the Thompson-Okanagan Plateau of the North American Deserts ecoregion. The Thompson-Okanagan Plateau has a dry continental climate and is in the rain-shadow of the Coast and Cascade Mountain Ranges; however, the region supports a strong agricultural industry that has a high irrigation demand, which accounts for 75% of the consumptive water use (Neilsen et al., 2006). 10 Tributary streams, like Whitman Creek, are the main source of water for the Okanagan Valley, and most streams in the Okanagan are fully allocated, with no leeway for further allocations (Cohen and Kulkarni, 2001).

The Capilano River headwater catchment is the warmest catchment in this study and has a mean annual temperature of 5.9°C. It is located in the Pacific and Nass Ranges of the Marine West Coast Forests ecoregion. The Pacific and Nass Ranges have a wet maritime climate, and the headwater catchment in this study has a mean annual precipitation of more than 200 cm. 15 The headwaters of the Capilano River fill the Capilano Reservoir, which supplies one-third of the water supply for the 2.5 million Metro Vancouver residents (Metro Vancouver, 2017).

**Table 1.** Catchment characteristics, including baseline 1980s (1970-1999) mean annual precipitation (P), mean annual temperature (T), and mean winter (1-Nov to 1-Apr) temperature ($T_w$).

| Level I Ecoregion | Watershed | Lat. | Lon. | Area [km²] | Elevation [masl][a] | Slope | P [cm] | T [°C] | $T_w$ [°C] |
|---|---|---|---|---|---|---|---|---|---|
| Taiga | Fort Nelson River | 58.5 | -123.0 | 7.5 | 564 | 2° | 45.9 | -0.6 | -15.1 |
| Northern Forests | Blueberry River | 57.0 | -121.9 | 3.2 | 935 | 3° | 49.8 | 0.0 | -11.7 |
| N American Deserts | Whiteman Creek | 50.2 | -119.7 | 7.2 | 1572 | 10° | 65.0 | 2.1 | -6.1 |
| Marine W Coast Forests | Capilano River | 49.5 | -123.2 | 4.5 | 1320 | 35° | 234.6 | 5.9 | 0.0 |

20      [a] masl: metres above sea level




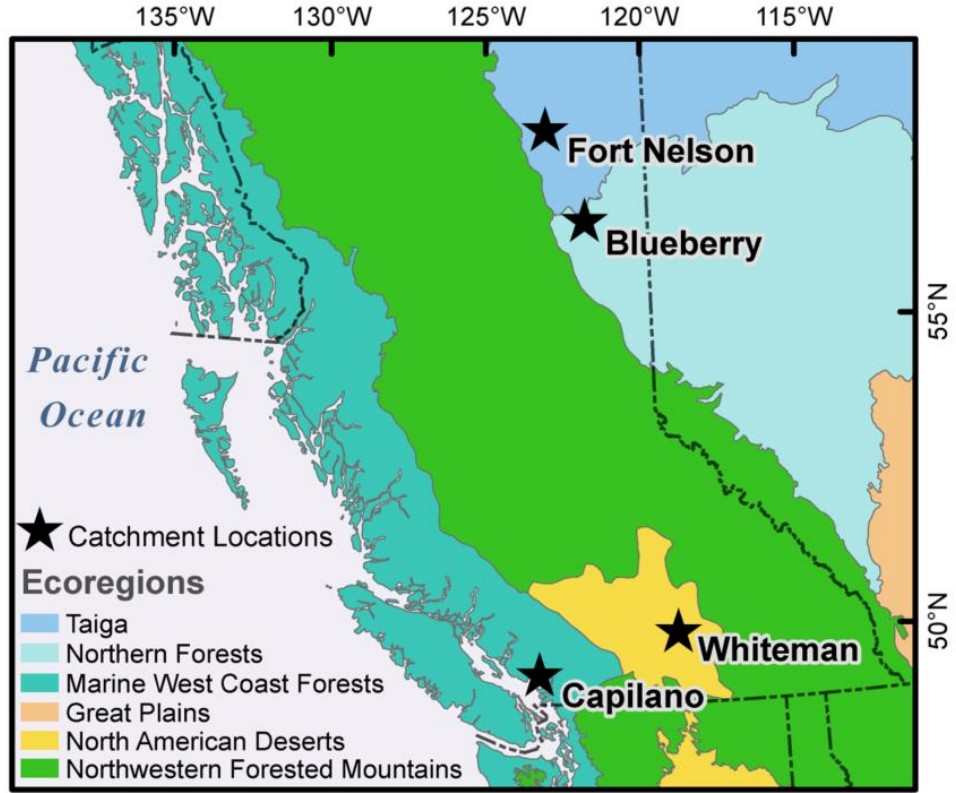

**Figure 1.** Headwater catchment locations and Level I ecoregions (CEC, 2011) in British Columbia, Canada.

## 3 Methods

The following sections describe the groundwater-surface water (GW-SW) models that were developed for the study
(Sect. 3.1) and the methodology used to assess impacts on the snow drought and low flow regimes (Sect. 3.2).

### 3.1 Groundwater-surface water (GW-SW) modelling

Groundwater discharge during low flow and drought periods is dependent on the amount of snow that accumulates
during winter and on the timing and rate of snowmelt and resulting groundwater recharge in the spring and summer (Tague
and Grant, 2009; Godsey et al., 2014; Meixner et al., 2016). Therefore, a comprehensive approach for analyzing climate change
impacts on snow drought and streamflow drought requires the application of a distributed, physically-based groundwater-
surface water model. With this type of model, parameters are directly related to the physical characteristics of the catchment.
Compared to the two other main types of hydrological models, i.e. empirical models and lumped conceptual models, distributed
physically based models are more appropriate for simulating ungauged catchments (Refsgaard and Knudsen, 1996) and for
use where significant changes in catchment conditions are expected (Klemes, 1985; Refsgaard and Knudsen, 1996), e.g.,
climate change scenario modelling. Therefore, this study uses the distributed physically based GW-SW modelling code MIKE





SHE – MIKE 11 (Danish Hydraulic Institute [DHI], 2007). MIKE SHE has been used in previous climate change scenario modelling studies, including studies in catchments with seasonal snow cover (Liu et al., 2011; Thompson, 2012; Foster and Allen, 2015), and has been compared to other modelling codes and shown to adequately model stream discharge (Vansteenkiste et al., 2013; Golmohammadi et al., 2014) and groundwater recharge (Foster and Allen, 2015).

MIKE SHE is a fully distributed hydrologic modelling code that can simulate actual evapotranspiration (AET), overland flow, one-dimensional (1D) unsaturated flow, and three-dimensional (3D) variably saturated groundwater flow. Rivers, lakes, and other channels are simulated by the MIKE 11 model, which is coupled to the MIKE SHE model with river links ($h$-points). Further details on the comprehensive modelling capabilities of the MIKE SHE software can be found in the user manual (i.e. DHI, 2007).

Model boundary conditions were consistent between all models and consisted of:

1.  zero flux boundaries at the catchment boundaries – representing topographical divides;

2.  zero flux boundaries at the bottom of the saturated bedrock layer (200 metres below ground surface);

3.  closed (zero flux) boundaries at the upstream end of the stream network branches; and

4.  an open (head-dependent flux) boundary defined by a discharge-elevation (Q-h) rating curve at the
downstream outflow.

These boundary conditions direct the input (precipitation) onto the model domain and partition it into two outputs: 1) evapotranspiration or 2) surface water flow through the downstream flux boundary. Initial groundwater levels were assigned to coincide with the ground surface and declined to dynamically stable levels during model spin up. The models were run for 150-year periods (1950-2100), using the first 20 years (1950-1969) as the spin up period to achieve a dynamically stable
state.

The GW-SW models in this study were developed to explore the relationship between snow drought and hydrological drought in idealized systems. Each of these small, ungauged headwater catchment models represent a different ecoregion in British Columbia. While real-world topography, stream networks, etc. were used, all models in this study are generic and were not put through any calibration or validation procedure; therefore, these models represent interpretive tools
(Anderson et al., 2015). The modelling code used in this study meets the guidelines laid out by Freeze and Harlan (1969) for adequate physics-based hydrological modelling and the climate change application criteria of Klemes (1985). Beven (1989) and Grayson et al. (1992), however, warn against the overparameterization of physically based models. This study aimed to represent the hydrologic systems in the simplest way possible and used homogenous land cover, soils, and geology for each catchment. Additional details on the model setups, including land surface, saturated zone, and unsaturated zone parameters
and stream network data are included in the following sections.




### 3.1.1 Land surface data and overland (OL) flow

Actual transpiration and soil evaporation were calculated using the equations from Kristensen and Jensen (1975). Required inputs include leaf area index (LAI), canopy interception, root characteristics, and empirical coefficients. Leaf area index (LAI) for each catchment was estimated using the 10-day interval LAI dataset from Gonsamo and Chen (2014). The required root characteristics include root depth and a root mass distribution parameter Aroot. Aroot was left at the default value (0.25 m$^{-1}$; DHI, 2007). Root depth was set at 400 mm for all catchments based on the findings of Curt et al. (2001). The canopy interception parameter $C_{int}$ and the empirical coefficients C1, C2, C3 were left at the default values (0.05 mm and 0.3, 0.2, and 20 mm/day, respectively; DHI, 2007), following Voeckler et al. (2014) and Foster and Allen (2015).

Overland flow occurs via two mechanisms: 1) exceedance of the soil infiltration capacity or 2) water table intersects the ground surface and generates return flow. Within MIKE SHE, overland flow is routed by surface topography, with the rate dependent on the diffusive wave approximation of the Saint Venant equation. Topography for all models was assigned using the Canadian Digital Elevation Model (CDEM; Natural Resources Canada, 2017). Land cover data (DataBC, 2013) were used to define surface roughness values (Manning's $M$), which control the resistance to overland flow. Manning's $M$ is the reciprocal of Manning's $n$; therefore, Manning's $n$ values from Chow (1959) were used to estimate Manning's $M$ values (Table S1).

### 3.1.2 Unsaturated and saturated zone

The unsaturated zone (UZ) within MIKE SHE is the zone through which the water table rises and falls. Vertical flow in the UZ was modelled using Richards' equation (Richards, 1931). Three-dimensional groundwater flow in the saturated zone (SZ) is based on Darcy's equation and is solved implicitly using a finite difference technique. The UZ and SZ are explicitly coupled, and the upper boundary of the SZ is a flux boundary which receives recharge from the SZ. Flux from the unsaturated to the saturated zone varies in time and is computed at the interface of the two zones. Because of the coupling between the two zones, the UZ and SZ must overlap, with the UZ extending to a depth of the lowest possible groundwater head. Depth to groundwater in the catchments is unknown; therefore, to ensure coupling between the UZ and SZ zones, the UZ and SZ zones were assigned the same number of layers, with the same depths, and the same hydraulic properties, extending from ground surface to 200 metres depth.

UZ and SZ layer depths, bulk densities, and vertical and horizontal saturated hydraulic conductivities ($K_z$, $K_{xy}$) were assigned based on British Columbia (BC) soil descriptions (Agriculture and Agri-Food Canada, 2013). Textural classes were determined from the BC soil descriptions and used to assign values of effective porosity ($\theta_s$), residual porosity ($\theta_r$), and empirical constants $\alpha$ and $n$ based on values in Carsel and Parrish (1988). Specific yield values were estimated from Morris and Johnson (1967). Parameters for the organic soil layers in the Fort Nelson (Taiga ecoregion) catchment were based on values from Letts et al. (2000). Soil names, bedrock geology, and the associated parameters are provided in Tables S2 to S5.





### 3.1.3 MIKE 11 stream network

Stream routing was modelled within MIKE 11 and requires four main components: a stream network, cross-sections, boundary conditions, and hydrodynamic parameters. Stream networks and drainage boundaries were obtained from British Columbia Freshwater Atlas (British Columbia, 2013). Stream cross-sections were digitized from surface topography (i.e. from the CDEMs). For each catchment, stream network boundary conditions consisted of closed (zero-flux) boundaries at the upstream ends of the stream network branches and an open (head-dependent) flux boundary at the downstream outflow. Rating curves for the downstream head-dependent flux boundaries were calculated using Manning's equation, shown in Eq. (1).

$$Q \ = A \ \times \frac{1}{n} \times R_h^{\frac{2}{3}} \times \sqrt{S} \qquad\qquad (1)$$

where $Q$ [m³/s] is the discharge leaving the model domain, $A$ [m²] is the cross-sectional area, $n$ is Manning's roughness coefficient, $R$ is the hydraulic radius [m], and S [m/m] is the channel slope. For each catchment, the channel slope was calculated from the CDEM and the hydraulic radius was calculated from the downstream cross-section for a range of possible stream stages spanning low flow to high flow conditions. A value of $Q$ was determined for each stage value using Eq. (1), thereby creating the rating curve required for the head-dependent flux boundary.

The global bed resistance Manning's roughness coefficient (Manning's $n$) was set to 0.05 for all catchments based on values in Chow (1959). Conductance values, which control water flow between the stream network and the saturated zone, were estimated from the vertical hydraulic conductivity values of the upper soil horizons (Tables S2-S5).

### 3.1.4 Climate change scenario modelling

Statistically downscaled climate projections from three global climate models (GCMs) from Phase 5 of the Coupled Model Intercomparison Project (CMIP5) were used to assess climate change impacts on low flows and snow drought. Two emissions pathways, representative concentration pathways (RCPs) 4.5 and 8.5 were selected – these represent different trajectories of anthropogenic radiative forcing, leading to radiative forcing levels of 4.5 and 8.5 W/m² by the end of 21st century (van Vuuren et al., 2011). RCP 4.5 represents a medium stabilization scenario, and RCP 8.5 represents a very high baseline emissions scenario (van Vuuren et al., 2011). A subset of three GCMs (CNRM-CM5-1, CanESM2-r1, ACCESS1-0-r1) was selected following recommendations in Cannon (2015) with the goal of capturing the widest spread in possible future climate. Daily climate time series (maximum temperature, minimum temperature, and precipitation) statistically downscaled with bias-correction/constructed analogues with quantile mapping reordering (BCCAQ) were downloaded from the Pacific Climate Impacts Consortium (PCIC) data portal (PCIC, 2014) covering the period of 1950 to 2100. Werner and Cannon (2016) showed that, out of the seven downscaling methods tested, BCCAQ performed best for reproducing hydrologically relevant climate extremes. Mean daily temperature was calculated as the average of the minimum and maximum daily temperature and used as the input for MIKE SHE. A comparison of climate data between the baseline 1980s (1970-1999) and two future periods, 2050s (2041-2069) and 2080s (2070-2099) is included in results (Sect. 4.1).





### 3.1.5 Evapotranspiration

In addition to mean daily temperature and daily precipitation, MIKE SHE requires estimates of potential evapotranspiration (PET). Potential evapotranspiration (PET) was calculated with the FAO Penman-Monteith method (Allen et al., 1998) using the R package "sirad" (Bojanowski, 2016). Daily solar radiation inputs for the Penman-Monteith method were estimated from daily maximum and minimum temperature using the Hargreaves and Samani (1985) model, following recommendations in Aladenola and Madramootoo (2012). Estimates of daily mean wind speed were unavailable, and a constant wind speed of 5 km/hr was used for all PET calculations, which is within the range of climate normals for the nearby climate stations (Environment and Climate Change Canada, 2018). Wind speed exhibits relatively minor impacts on PET (McKenney and Rosenberg, 1993; Gong et al., 2006; Tabari and Talaee, 2014; Córdova et al., 2015); therefore, the use of a constant wind speed was deemed acceptable.

### 3.1.6 Snow

Within MIKE SHE, snow accumulation and melt are modelled using a threshold melting temperature, a maximum wet snow storage fraction, and a degree-day coefficient. The threshold melting temperature for all catchments was set to 0°C, and the maximum wet snow storage fraction was set to 0.2, which is in the mid-range of values used in previous studies (Wijesekara et al., 2014; Voeckler et al., 2014; Foster and Allen, 2015). A value of 2.74 mm/degree-day C was used for the degree-day coefficient in all models based on recommendations in United States Department of Agriculture (USDA) National Engineering Handbook (Van Mullem and Garen, 2004). The minimum snow storage was set to 0 mm for all catchments.

This snowmelt methodology, referred to as the temperature-index or degree-day method, assumes an empirical relationship between air temperatures and melt rates and has been widely applied due to its simplicity (e.g., Clyde, 1931; Corps of Engineers, 1956; World Meteorological Organization [WMO], 1986; Van Mullem and Garen, 2004). The degree-day method does not, however, account for several factors that are important for snowmelt, including wind speed, humidity, topography (slope, aspect, and shading), cloud cover, and vegetation (Male and Granger, 1981; Gray and Landine, 1988; Harding and Pomeroy, 1996; Pomeroy et al., 1998; Marks et al., 1999; among others). Despite the over-simplification and documented short-comings of this method, temperature-index methods often perform well at the catchment scale (World Meteorological Organization [WMO], 1986; Sand, 1992; Rango and Martinec, 1995; Hock, 2003) and can match the performance of energy balance models (WMO, 1986).

To evaluate the ability of the degree-day method to capture the response of snow processes to climate change, a supplemental analysis was completed, comparing the degree-day method used in MIKE-SHE to the more complex energy-balance method (Gray and Landine, 1987; Pomeroy et al., 2007) used in the Cold Regions Hydrological Model (CRHM; Pomeroy et al., 2012). MIKE SHE and CRHM each have an extensive record of being used in climate change studies that have included extensive validation and confirmation with observations (MIKE SHE: Thompson, 2012; Vansteenkiste et al., 2013;



Golmohammadi et al., 2014; CRHM: López-Moreno et al., 2013; Rasouli et al., 2014, 2019; Harder et al., 2015; Weber et al., 2016; Krogh and Pomeroy, 2018). Further, CRHM has a proven ability to model snow accumulation and melt in both prairie and alpine basins (Fang and Pomeroy, 2007; Pomeroy et al., 2007; Pomeroy et al., 2012) – even in data scarce regions where calibration is not possible (Pomeroy et al., 2007). As a semi-distributed hydrological modelling code, CRHM does not include
physically-based groundwater flow and thus would not be appropriate for investigating the relationship between snow and streamflow drought. The comparison of MIKE SHE to CRHM serves as a test of the validity of the SWE outputs from MIKE SHE in these data scarce headwater catchments.

The comparison between MIKE SHE's degree-day method and CRHM's energy-balance method was completed for two of the four catchments: 1) Capilano and 2) Blueberry. The two methods exhibit greater divergence at temperatures near
the rain-snow transition (i.e. in the warmer Capilano catchment) than at temperatures below the rain-snow transition (i.e. in the Blueberry catchment) (Figure 2). Further, the energy-balance method results in large differences between landcover types, unlike the degree-day method which exhibits no difference between landcover types (Figure 2). The emphasis of this study, however, is on the sensitivity of snow hydrology to climate change, which is analyzed in terms of relative change and not absolute change. Thus, despite differences in the absolute values, the relationship between the baseline and future simulations
is the same for both methods – a continuing decline in winter snowpack. While further study using energy balance models should be completed to investigate within-catchment spatial differences in the response of snow drought regimes to climate warming, MIKE SHE's degree-day method was deemed sufficient for this combined investigation of snow drought, streamflow drought, and groundwater recharge in these generic headwater catchment models.



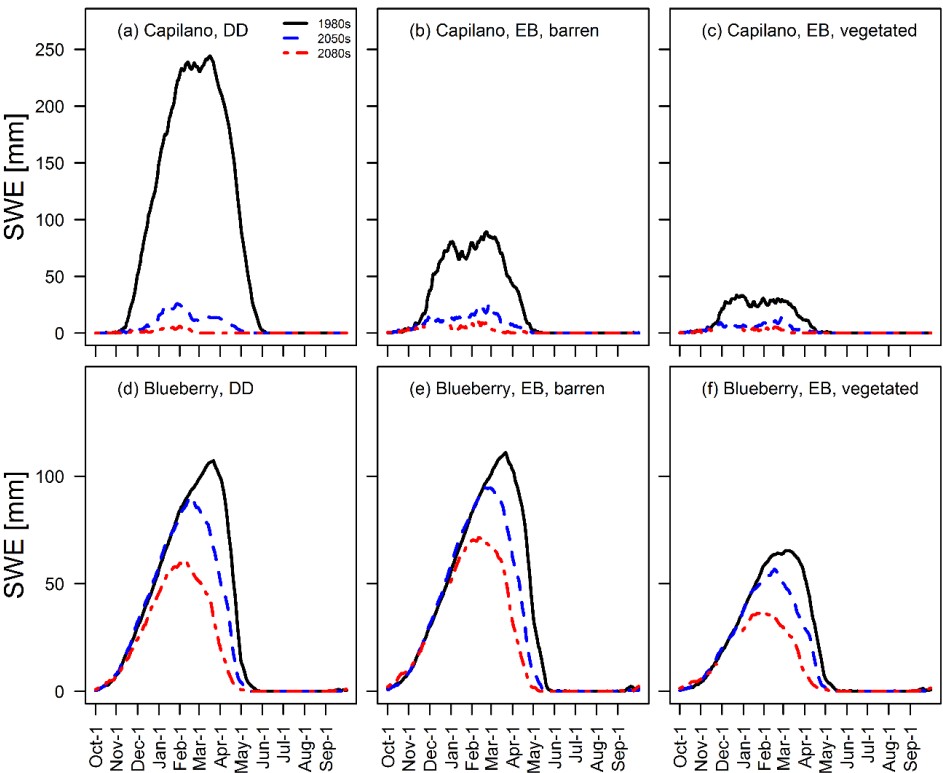

**Figure 2.** Simulated snow water equivalent (SWE) comparison between degree-day (DD) method (a & d), using MIKE-SHE, and an energy-balance (EB) method (b, c, e, f), using the Cold Regions Hydrological Model (CRHM). Plots show mean daily SWE for 1980s baseline (1970-1999), 2050s (2040-2069), and 2080s (2070-2099) for representative concentration pathway (RCP) 8.5 for two different catchments, Capilano (a, b, c) and Blueberry (c, d, e). For the EB method, results from two different landcovers are shown: 1) barren (b, e) and 2) vegetated (c, f). Results from different landcovers are not shown for the DD method because landcover has no effect, under any landcover, the results would be the same as (a) and (d).

### 3.2 Analysis methods

#### 3.2.1 Snow drought

To investigate how different snow drought types impact seasonal low flows, snow droughts were classified using the methodology outlined in Dierauer et al. (2019). With this method, winters with below-normal peak snow water equivalent (SWE) are classified as warm, dry, or warm and dry snow droughts based on winter precipitation ($P_w$) and winter thawing degrees ($TD_w$). Years with below-normal peak SWE, below-normal $P_w$, and below-normal $TD_w$ are classified as "dry" snow droughts; years with below-normal peak SWE and above-normal $P_w$ are classified as "warm" snow droughts; and years with below-normal peak SWE, below-normal $P_w$, and above-normal $TD_w$ are classified as "warm and dry" snow droughts. Peak SWE, $P_w$, and $TD_w$ normals were defined using the baseline 1980s period (1970-1999). For each catchment, a simplified winter versus summer seasonal classification was defined using the 1980s baseline 25th percentile of mean daily temperature ($\bar{T}_{25}$), with days of the year with $\bar{T}_{25} < 0°C$ corresponding to "winter" and days of the year with $\bar{T}_{25} \geq 0°C$ corresponding to





"summer". The hydrologic year start was defined by the start of the "winter" season and held constant through all years and all climate change scenarios. After classification, severity (mean deficit below baseline normal), frequency (fraction of years), and risk (frequency x severity) were calculated for each snow drought type.

### 3.2.2 Low flows and streamflow drought

Low flows are generated by different hydrological processes in winter versus summer, i.e. below freezing temperatures and snow accumulation versus above freezing temperatures and high evapotranspiration rates (Waylen and Woo, 1987; Laaha and Bloschl, 2006; Burn et al., 2008). Therefore, the same simplified seasonal classification outlined in Sect. 3.2.1 was applied to the low flow analysis. The magnitude of low flows was quantified using two metrics: 1) mean 15-day minimum runoff (MAM15) and 2) mean 30-day minimum runoff (MAM30). The two metrics were calculated separately for
the summer and winter season, indicated by "s" and "w" subscripts, respectively (Table 2).

       To evaluate the propagation of snow drought into summer streamflow drought, the frequency and mean severity of summer streamflow droughts following a snow drought were tabulated, considering only years with summer precipitation above the 1980s baseline normal, where "summer" is defined as in Sect. 3.2.1. By doing this, summer streamflow droughts caused by a summer precipitation deficit could be separated from summer streamflow droughts caused by snow droughts, and
thus the propagation of snow drought into summer streamflow drought could be highlighted. Summer streamflow drought severity was analysed using the summer low flow metrics, identifying low flows of anomalously low magnitude based on an exceedance threshold, as indicated in Table 2. A baseline threshold of 80% exceedance frequency was used, i.e. low flow magnitude that was exceeded 80% of the time during the baseline period. Summer streamflow drought years were then defined as years with MAM15$_s$ and/or MAM30$_s$ values below the threshold.

**Table 2.** Low flow regime indicators, calculated yearly.

| Winter | Summer | Description | Units | Threshold |
|--------|--------|-------------|-------|-----------|
| MAM15$_w$ | MAM15$_s$ | Mean 15-day minimum runoff | mm/day | <80% exceedance |
| MAM30$_w$ | MAM30$_s$ | Mean 30-day minimum runoff | mm/day | <80% exceedance |

### 4. Results

       The following sections present and discuss the results of the GW-SW climate change scenario modelling, including climate change impacts on the annual and intra-annual water balance (Sect. 4.1) and impacts on snow drought and low flows
(Sect. 4.2 and Sect. 4.3).

### 4.1 Climate change impacts on the annual and intra-annual water balance

       A water balance analysis was completed for each GCM-RCP combination, for a total of six 150-year water balance time series for each headwater catchment. The water balance components of interest for this study include precipitation,





snow water equivalent (SWE), runoff, actual evapotranspiration (AET), and groundwater recharge. Groundwater recharge represents water that enters the SZ from the UZ; however, with the MIKE SHE water balance tool, both flux into and out of the SZ are tabulated. Thus, recharge can be positive (outward flux to UZ) or negative (inward flux to SZ). With no change in groundwater storage, the inward fluxes in recharge zones would be balanced by outward fluxes in discharge zones, and total

annual recharge would roughly equal 0. Therefore, a detailed saturated zone water balance was extracted for a small upland region of each catchment to analyse groundwater recharge in a recharge zone only. Runoff for each catchment was calculated as the volume of streamflow leaving the downstream head-dependent flux boundary, converted from m³/s to mm/day by dividing by catchment area.

        Water balance results for individual GCMs are not shown but rather lumped by RCP for simplicity. Results are

summarized for a baseline period (1980s) and two future periods (2050s and 2080s) and are presented as both the absolute change (future – baseline)(Figure S1) and relative change ([future – baseline] / baseline)(Figure 3, Table S6) for the three-member GCM ensemble. Average annual water balance errors for all models were less than 3%; however, due to transient conditions and changes in subsurface storage, mean annual AET plus mean annual runoff is not equal to mean annual precipitation.

15        The statistically downscaled climate change projections show increases in temperature and precipitation for all four catchments (Figure 3, Table S6). The relative increase in precipitation is highest in the two northern catchments (Fort Nelson and Blueberry; Figure 3), which have a cold, dry climate. The absolute increase (mm/year) in precipitation, however, is highest in the warmest, wettest catchment – Capilano (Figure S1). The seasonal distribution of precipitation does not change substantially under RCPs 4.5 and 8.5; however, in general, the fall (Sep to Nov) and spring (Mar to May) seasons exhibit the

largest relative increases in precipitation (Figure S2), and summer (Jun to Aug) and winter (Dec to Feb) exhibit the smallest. Compared to the projected changes in precipitation, changes in temperature are more similar among all catchments (Table S6). Projected temperature increases are highest in winter and lowest in fall (Figure S3). As expected, increases in the mean annual temperature are greatest for the high emissions scenario (RCP 8.5), and by the 2080s, mean annual temperature is projected to be 5.6 to 6.1°C higher compared to the baseline 1980s period (Table S6).

25        Annual runoff is projected to increase for all four catchments (). Relative (%) increases in runoff are largest for the coldest, driest catchment (Fort Nelson; Figure 3 and Table S6), while absolute (mm/year) increases in runoff are largest for the warmest, wettest catchment (Capilano; Figure S1). In addition to increases in annual runoff, the within-year distribution of runoff changes substantially. The spring freshet starts earlier for all catchments and decreases in magnitude for all but Fort Nelson (Figure S4), which is the northernmost and coldest catchment. In the warmest catchment (Capilano), the spring

freshet disappears completely for both future time periods under both RCPs. In all catchments, the slope of the spring freshet rising limb decreases, indicating a longer spring melt season with slower snowmelt (Figure S4). These changes (declined spring freshet peak and a longer melt season) are consistent with the findings of previous observation-based studies (Hamlet and Lettenmaier, 2007; Rood et al., 2008).





In general, changes in the timing and magnitude of the spring freshet are directly related to changes in the length of the snow-covered period and the magnitude and timing of peak SWE (Figure S5). While the increased annual precipitation leads to increased peak SWE for the coldest catchment (Fort Nelson), the impact of increased temperatures outweighs the impacts of increased precipitation in the remaining catchments, leading to no significant change in peak SWE (Blueberry) or

significant decreases in peak SWE (Whiteman and Capilano; Figure 3). The largest absolute and relative decreases in peak SWE occur in the warmest catchment, Capilano, which has a >90% decrease in peak SWE for the 2080s under RCP 8.5. In addition to changes in the magnitude of peak SWE, the average day of peak SWE and melt-out occur earlier, resulting in a shorter snow-covered period for all catchments for both future time periods under both RCPs (Figure S5). Additionally, snowmelt is slower for all catchments for both future periods, as illustrated by the shallower slope of the falling limbs

(Figure S5). The earlier and slower snowmelt is consistent with the Musselman et al. (2017) study, which showed that snow melts more slowly in a warmer world due to an increase in winter and spring melt and longer snow-free periods during times of high energy (i.e. summer).

As expected, the changes in temperature and precipitation, and associated changes in snow accumulation and melt, lead to significant changes in both the total annual AET (Figure 3) and the intra-annual patterns in AET (Figure S6). AET

increases significantly for all catchments except Whiteman, located in the Okanagan Valley, which exhibits no significant change in annual AET (Figure 3). Seasonally, AET increases most in late-winter and spring (Feb to May) and decreases (Whiteman and Capilano) or exhibits no substantial change (Fort Nelson and Blueberry) during summer (Figure S6). Decreased AET during the summer can be primarily attributed to the shift toward earlier snowmelt, which decreases summer water availability. This negative feedback between snowmelt timing and evapotranspiration has been discussed by Barnett et

al. (2005) and documented by previous climate change modelling studies (e.g., Shrestha et al., 2012). Shifts in vegetation patterns will likely influence catchment response to climate change (Alo and Wang, 2008; Teutschbein et al., 2018); however, it is difficult to project and constrain possible vegetation shifts, and vegetation change was not included in the modelling efforts. Changes in wind speed, either due to climate change or vegetation change, would also impact AET, an effect which was not considered in this study.

Within MIKE SHE, water that reaches the saturated zone (i.e., groundwater recharge) may then exit the saturated zone via evapotranspiration, baseflow to the river, or surface return. Groundwater recharge may be higher or lower than runoff, depending on catchment's physical properties (e.g., soils, geology, vegetation, ground slope) which control the evapotranspiration dynamics and the magnitude of overland flow. In the Fort Nelson catchment, which has high porosity organic soils (Table S2) and shallow slope (Table 1), groundwater recharge is much higher than runoff (Table S6), and a

large proportion of the water that reaches the saturated zone then leaves the system through evapotranspiration. In the Capilano catchment, which has lower porosity soils (Table S5) and steep slopes (Table 1), groundwater recharge is lower than runoff (Table S6) and a substantial portion of runoff is generated from overland flow.

At the annual time scale, recharge increases significantly for all but the warmest catchment (Capilano; Figure 3). Intra-annually, the patterns in groundwater recharge are primarily affected by changes in the onset of snow accumulation and





melt. In all catchments, the spring recharge peak starts earlier in the year and decreases in magnitude (Figure S7), resulting in higher winter groundwater storage, an earlier start to the spring/summer groundwater recession period, and thus decreased summer groundwater storage (Figure S8). Increased winter-season recharge for regions with seasonal snow cover is consistent with the results of previous climate change modelling studies (e.g., Eckhardt and Ulbrich, 2003; Jyrkama and

5  Sykes, 2007; Kovalevskii, 2007). A shift toward more rain and less snow in combination with slower snowmelt would suggest an overall decrease in groundwater recharge (Earman et al., 2006; Barnhart et al., 2016). While it is difficult to separate the effects of increased temperatures from the effects of increased precipitation, the results of this study show an increasing ratio of recharge to precipitation (R:P ratio) for the Fort Nelson catchment, relatively constant R:P ratios for the Blueberry and Whiteman catchments, and a decreasing R:P ratio for the Capilano catchment (Table S6). The different

10  responses of the R:P ratio (increase, no change, decrease) seem to be related to the catchment's starting point (in terms of temperature), as the coldest catchment exhibits an increase in the R:P ratio and the warmest catchment exhibits a decrease in the R:P ratio.

**Figure 3.** Relative change in annual climate and water balance components for the 1980s baseline (1970-1999) versus 2050s (2040-2069) and 2080s (2070-2099) for representative concentration pathway (RCP) 4.5 and RCP 8.5, including mean annual temperature (Temp), annual precipitation (Precip), peak snow water equivalent (SWE), annual runoff, annual actual evapotranspiration (AET), and annual groundwater recharge. Blue and orange shading indicate a significant ($p < 0.05$) increase or decrease relative to the baseline period, as assessed with the two-sided Mann-Whitney U test. Arrows are added for clarity where boxplot shading is unclear. Figure S1 shows the same data, plotted as absolute values, and Table S6 provides the corresponding mean annual values along with the absolute and relative change.





## 4.2 Snow drought

In response to the increased future precipitation and temperature (Figure 3), the snow drought regime changes substantially for all catchments. Warm snow droughts increase in frequency, and dry snow droughts decrease in frequency (Figure 4, Table S7). Additionally, warm, and warm and dry, snow drought severity increases for the two warmest catchments, Whiteman and Capilano (Figure 5, Table S8). In general, dry snow droughts transition to warm and dry snow droughts, and, by the 2080s, the frequency of dry snow drought drops to 0 for all catchments (Figure 4, Table S7). In terms of temperature, the magnitude of change in the snow drought regime is related to the catchment's starting point, with the warmest catchment (Capilano) exhibiting the largest increase in the frequency and severity of snow drought and the coldest catchment (Fort Nelson) exhibiting no substantial increase in the frequency or severity of snow drought (Figures 4 and 5).

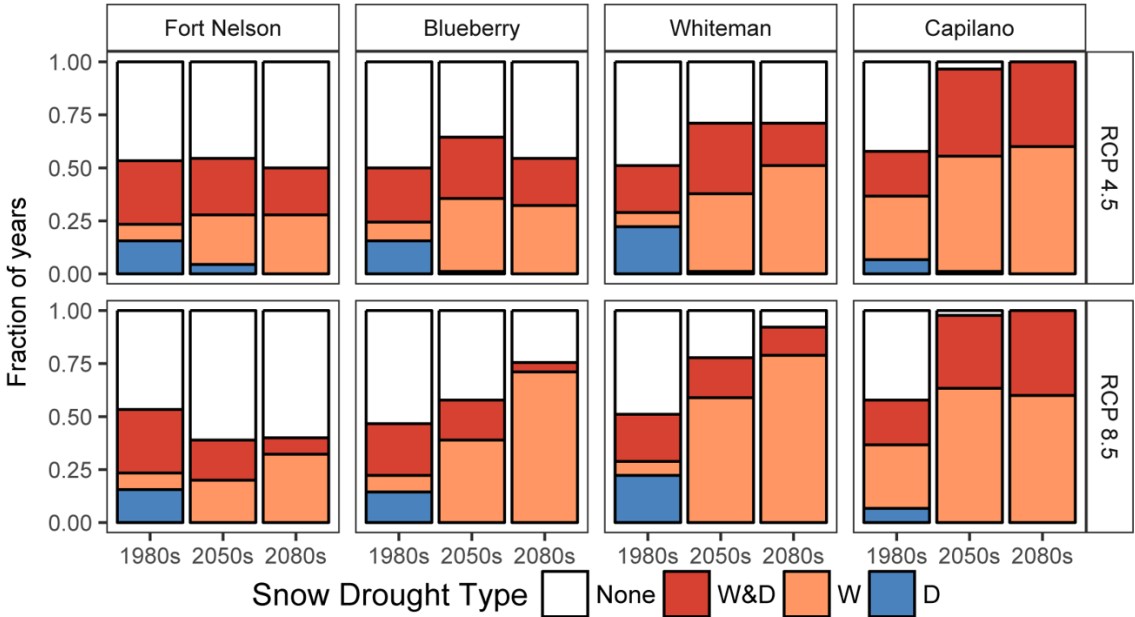

**Figure 4.** Frequency (fraction of years) of warm (W), dry (D), and warm and dry (W&D) snow droughts for the baseline 1980s (1970-1999) versus 2050s (2040-2069) and 2080s (2070-2099) for representative concentration pathway (RCP) 4.5 and RCP 8.5. Table S7 provides the same data in tabular format.





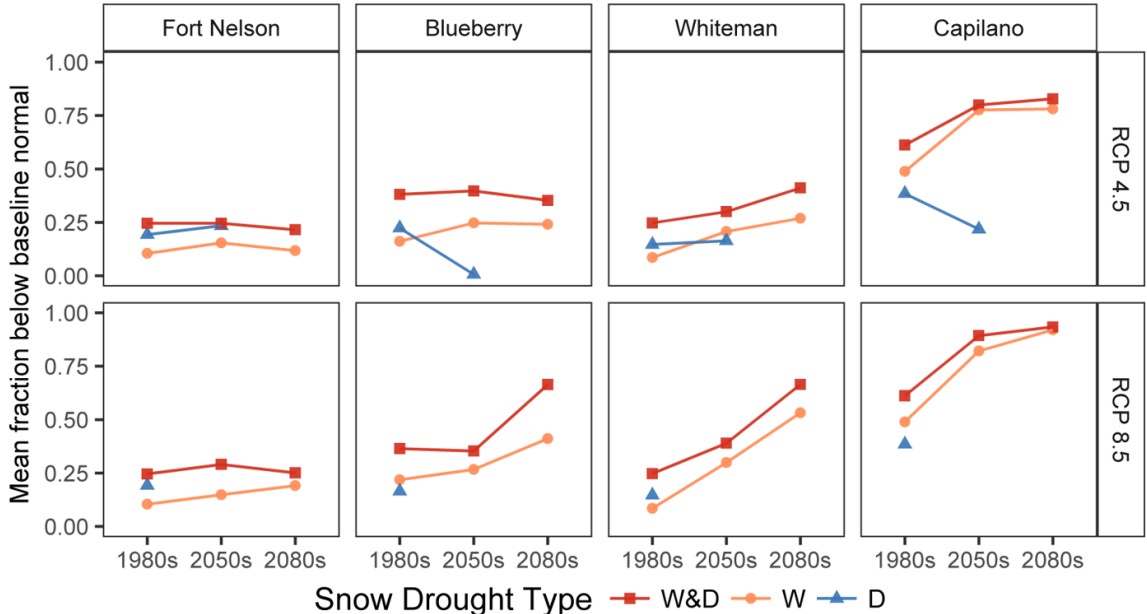

**Figure 5.** Mean severity (fraction below baseline normal) of warm (W), dry (D), and warm and dry (W&D) snow droughts for the baseline 1980s versus 2050s (2040-2069) and 2080s (2070-2099) for representative concentration pathway (RCP) 4.5 and RCP 8.5. Table S8 provides the same data in tabular format. Note: Dry snow droughts transition to warm and dry snow droughts and therefore have no mean severity plotted in some future time periods.

The increased frequency and severity of snow drought necessarily leads to increased snow drought risk, and overall, the changes in snow drought risk (Table 3) mirror the changes in snow drought severity (Figure 5). In general, snow drought regimes in all catchments shift toward more frequent, higher severity warm, and warm and dry, snow droughts, and less frequent, lower severity dry snow droughts. As documented by Dierauer et al. (2019) and shown in Figure S9, the response of warm snow drought risk to increased winter temperature is non-linear. A 2°C increase in the mean winter (1-Nov to 1-Apr) temperature corresponds to a substantially larger increase in warm snow drought risk for the Capilano catchment as compared to the Fort Nelson catchment. The two warmest catchments, Whiteman and Capilano, exhibit the largest increases in total snow drought risk. Due to the transition of dry snow droughts to warm and dry snow droughts, dry snow drought risk decreases for all catchments for both future time periods under both RCPs. The coldest catchment, Fort Nelson, exhibits a slight decrease in total snow drought risk.



**Table 3.** Risk (severity x frequency) for dry (D), warm (W), and warm and dry (W&D) snow droughts. Baseline 1980s (1970-1999) versus 2050s (2040-2069) and 2080s (2070-2099) for representative concentration pathways (RCPs) 4.5 and 8.5.

| | | Fort Nelson | | Blueberry | | Whiteman | | Capilano | |
|---|---|---|---|---|---|---|---|---|---|
| **1980s** | D | 3.0% | | 3.5% | | 3.2% | | 2.6% | |
| | W | 0.8% | | 1.4% | | 0.6% | | 14.6% | |
| | W&D | 7.4% | | 9.7% | | 5.5% | | 12.9% | |
| | *Total* | *11.2%* | | *14.6%* | | *9.3%* | | *30.1%* | |
| | *RCP* | *4.5* | *8.5* | *4.5* | *8.5* | *4.5* | *8.5* | *4.5* | *8.5* |
| **2050s** | D | 1.0% | -- | <0.1% | -- | 0.2% | -- | 0.2% | -- |
| | W | 3.6% | 3.0% | 8.5% | 10.4% | 7.6% | 17.6% | 42.2% | 52.0% |
| | W&D | 6.5% | 5.5% | 11.5% | 6.7% | 10.0% | 7.4% | 32.9% | 30.8% |
| | *Total* | *11.1%* | *8.5%* | *20%* | *17.1%* | *17.8%* | *25.0%* | *75.5%* | *82.8%* |
| **2080s** | D | -- | -- | -- | -- | -- | -- | -- | -- |
| | W | 3.3% | 6.1% | 7.8% | 29.2% | 13.8% | 41.9% | 46.9% | 55.2% |
| | W&D | 4.8% | 1.9% | 7.8% | 3.0% | 8.2% | 8.9% | 33.1% | 37.4% |
| | *Total* | *8.1%* | *8.0%* | *15.6%* | *32.2%* | *22.0%* | *50.8%* | *80.0%* | *92.6%* |

### 4.3 Low flows and summer streamflow droughts

As the snow drought regime shifts toward more frequent, higher severity temperature-related (i.e. warm, and warm and dry) snow droughts, the streamflow regime shifts toward less severe winter low flows and more severe summer low flows (Figure 6). Low flows are a normal feature of the natural flow regime (Smakhtin, 2001); however, anomalously low flows are

equivalent to streamflow droughts. Thus, a shift toward more severe (i.e. lower magnitude) summer low flows represents an increase in summer streamflow drought severity, and a shift toward less severe (i.e. higher magnitude) winter low flows represents a decrease in winter streamflow drought severity.

The impact of snow drought on summer and winter low flows is dependent on the snow drought type. Consistent with findings of Dierauer et al. (2018), warm, and warm and dry, snow droughts lead to more severe summer low flows and

significantly less severe winter low flows (Figure 7). In the context of climate warming and considering the relationship between snow drought and low flows shown in Figure 7, summer streamflow drought regimes are likely to shift toward more frequent, higher severity snow-drought related events. Using a threshold-based approach to define summer streamflow drought years for each low flow metric (see Sect. 3.2) shows that, in the absence of summer precipitation deficit, snow droughts propagate into summer streamflow droughts more frequently in the future time periods as compared to the baseline 1980s

(Figure 8). Thus, warm snow droughts not only become more frequent and severe in the future (Figures 4 and 5) but are also more likely to result in summer streamflow drought conditions. Dry snow droughts, on the other hand, become less frequent



in the future (Figure 4) and, in the absence of summer precipitation deficit, are unlikely to be followed by a summer streamflow droughts (Figure 8).

The warm snow season streamflow drought events identified in this study are strictly temperature-driven, as both winter and summer precipitation are above the baseline 1980s normal. Climate change impacts on the frequency of these

5  events vary between catchments due to the baseline winter air temperature. The Fort Nelson catchment, which has winter air temperatures far below zero, exhibits minimal increase in the occurrence of warm snow season streamflow droughts, while Capilano catchment, which has winter air temperatures near zero, exhibits a large increase in the occurrence of warm snow season streamflow droughts (Figure 8).

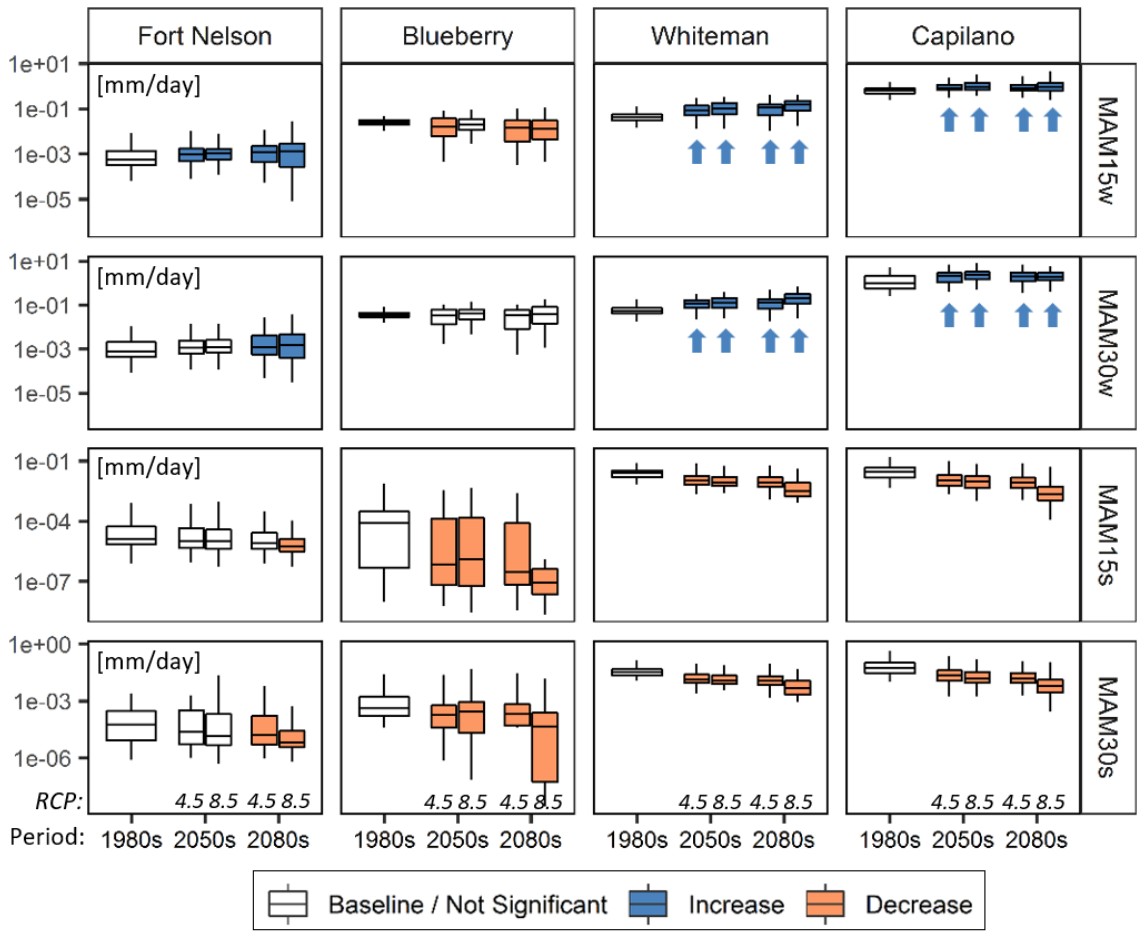

**Figure 6.** Low flow metrics for the 1980s baseline (1970-1999) versus 2050s (2040-2069) and 2080s (2070-2099) for representative concentration pathway (RCP) 4.5 and RCP 8.5. Blue and orange shading indicate a significant (p < 0.05) increase or decrease relative to the baseline period, as assessed with the two-sided Mann-Whitney U test. Abbreviations are as in Table 2.



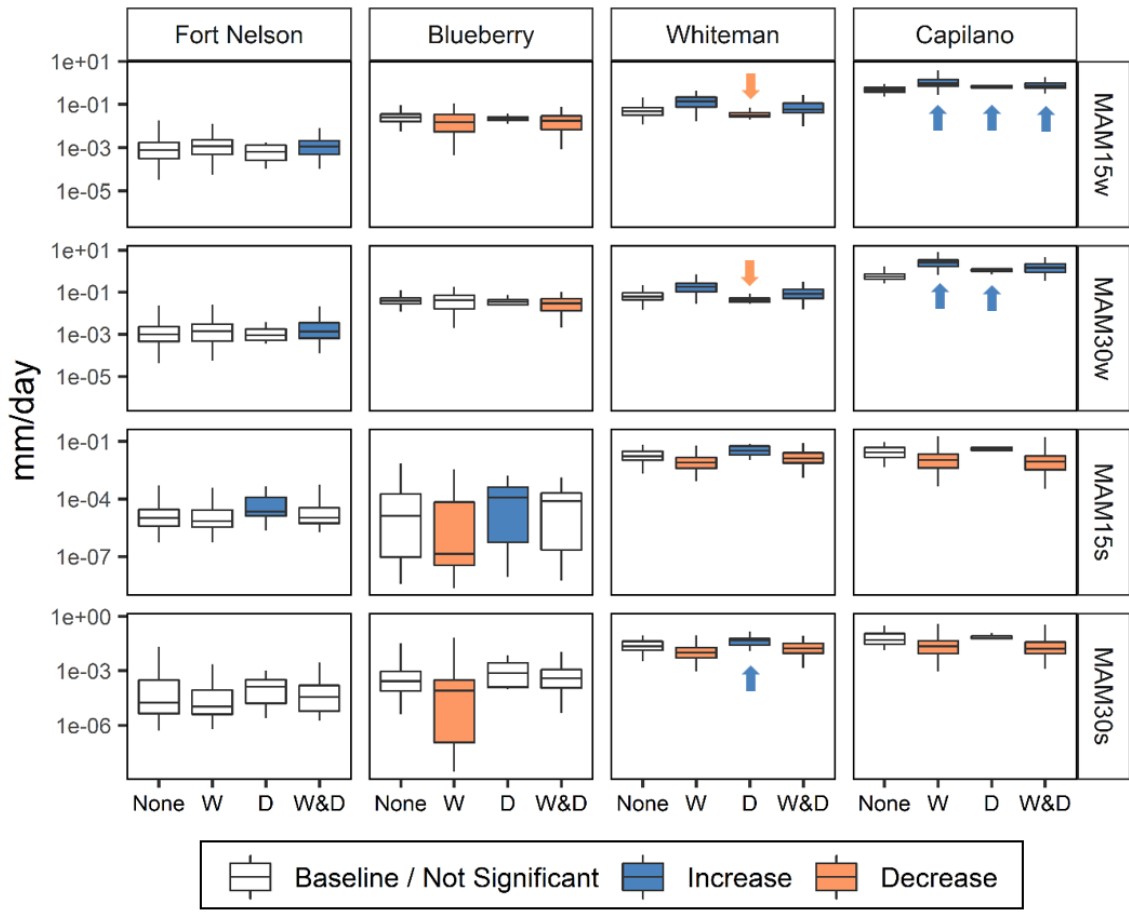

**Figure 7.** Snow drought impacts on low flows by snow drought type, including years without snow drought (None) and years with warm (W), dry (D), and warm and dry (W&D) snow droughts. Blue and orange shading indicate the values are significantly (p < 0.05) higher or lower relative to years without snow drought, as assessed with the two-sided Mann-Whitney U test. Abbreviations are as in Table 2. Arrows are added for clarity where boxplot shading is unclear.

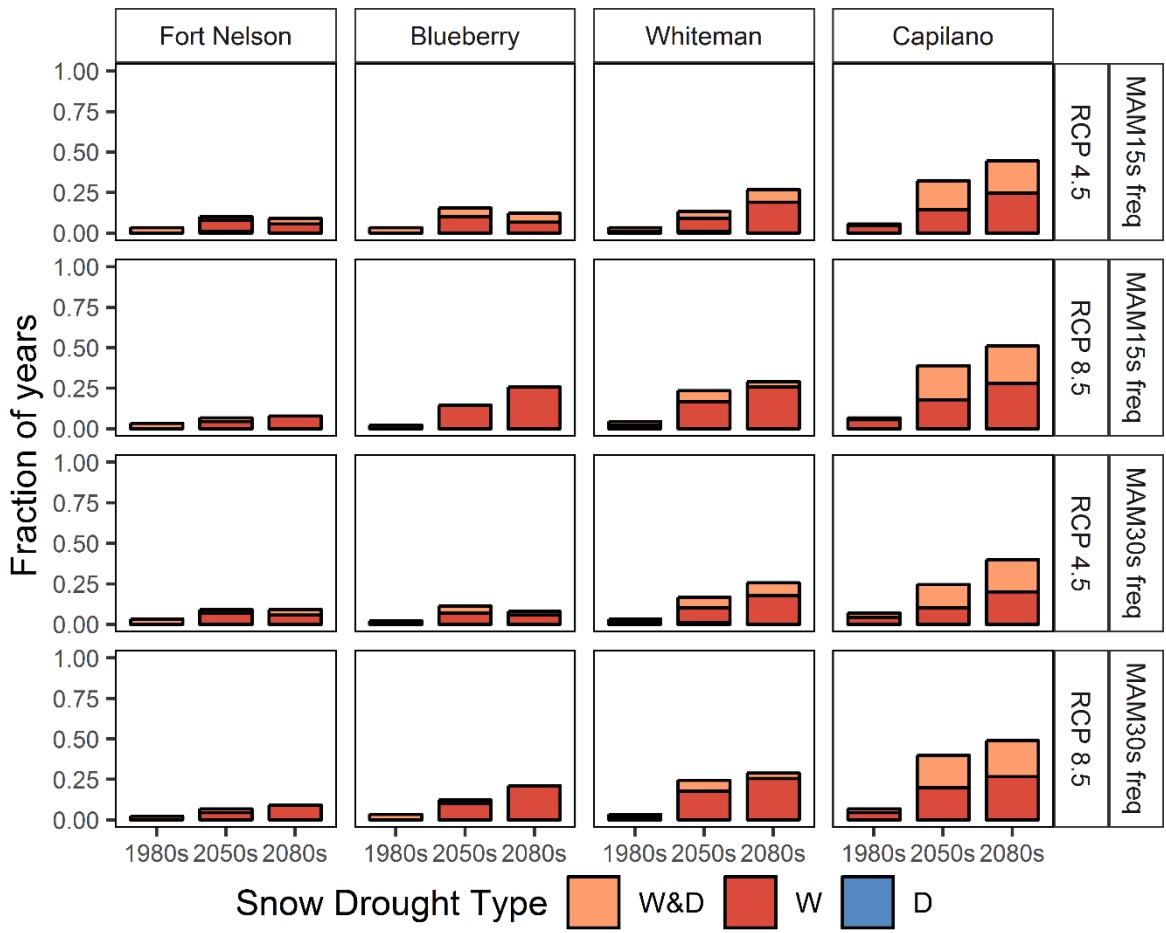

**Figure 8.** Frequency of snow drought propagation into summer streamflow drought, in the absence of summer precipitation deficit, by snow drought type: warm (W), Dry (D), warm and dry (W&D), for representative concentration pathway (RCP) 4.5 and 8.5. Abbreviations are as in Table 2.

## 5 Discussion

While several recent studies have focused of snow drought (Dierauer et al., 2019; Mote et al., 2016; Harpold et al., 2017) and its hydrological impacts (Cooper et al., 2016; Sproles et al., 2017; Hatchett and McEvoy, 2018), no previous studies have explicitly related climate change impacts on snow drought to subsequent impacts on summer low flows and summer streamflow drought. In this study, generic GW-SW models of headwater catchments were combined with downscaled climate change projections for two different RCPs. Climate change projections show increases in both precipitation and temperature, leading to decreases in the frequency and severity of dry snow droughts and increases in the frequency and severity of warm, and warm and dry, snow droughts. Climate warming and the subsequent shifts in the snow drought regime result in decreased summer runoff, decreased summer groundwater storage, and more severe summer low



flows. Climate warming has the opposite effect on the winter season, with model results showing increased winter runoff, increased winter groundwater storage, and less severe winter low flows.

Snow droughts have direct impacts on summer low flows (Figure 7), and temperature-related (i.e. warm, and warm and dry) snow droughts not only become more frequent and severe in the future but are also more likely to result in summer

streamflow drought conditions (Figure 8). The shift to lower severity winter streamflow droughts and higher severity summer streamflow droughts is consistent with the results of previous hydrologic modelling studies (Feyen and Dankers, 2009; Wanders and Van Lanen, 2015) and with the general hypothesis that streamflow droughts with different causative factors will respond differently to climate change (Van Loon and Van Lanen, 2012; Van Loon et al., 2015). In general, increased summer streamflow drought severity and decreased winter streamflow drought severity are consistent with an

overall shift in the intra-annual distribution of runoff, an impact of climate warming on snowmelt hydrology which has been documented by many previous studies (Leith and Whitfield, 1998; Whitfield and Cannon, 2000; Adam et al., 2009; Déry et al., 2009; Pederson et al., 2011; among others).

Consistent with the results of Dierauer et al. (2019), the response of snow drought risk to climate warming is non-linear (Figure S9), and the magnitude of change in the snow drought and low flow regime is related to the catchment's

baseline mean winter (1-Nov to 1-Apr) temperature. Because of the nonlinear relationship between temperature and snow drought risk, a +2°C change in the mean winter temperature has a larger impact on the snow drought regime in catchments with winter temperatures near zero (e.g., the Capilano catchment) compared to catchments with winter temperatures far below zero (e.g., the Fort Nelson catchment). Because of the impacts of snow drought on summer low flows, warmer catchments also exhibit greater increases in the severity of summer low flows compared to colder catchments.

The shift toward more frequent and more severe temperature-related snow droughts and longer, more severe summer low flow periods will have wide-ranging effects on terrestrial and aquatic ecosystems. In the terrestrial realm, earlier snow disappearance will lead to less water for mountain ecosystems (Harpold, 2016), lower carbon uptake (Hu et al., 2010; Winchell et al., 2016), more wildfires (Westerling et al., 2006), and more tree death (Bales et al., 2018). In the aquatic realm, climate warming coupled with the shift toward more severe summer low flows will have compound impacts on stream

temperature, which increases with air temperature and decreases with streamflow rates (Hockey et al., 1982; Webb et al., 2003). Summer low flows of sufficient magnitude are critical for aquatic ecosystem health (Fleming et al., 2007; Moore et al., 2013), and higher stream temperatures negatively impact species distributions and decrease growth rates (Beschta et al., 1987, Eaton and Scheller, 1996). Further, summer low flows that are lower than normal, i.e. drought conditions, reduce habitat availability (Lake, 2003), increase pollutant concentrations (Mosley, 2015), and lower the oxygen available to aquatic

organisms (Sprague, 2005; van Vliet and Zwolsman, 2008; Ylla et al., 2010).

Shifts in snow drought, low flows, and streamflow drought regimes will also have widespread implications for surface water supply security. Increased frequency of warm snow droughts will likely lead to an increased frequency of mid-winter melt events (Hatchett and McEvoy, 2018), which will create challenges for reservoir management. Winter melt events should be of low intensity (e.g., Musselman et al., 2017); however, climate change may also result in increased rain on snow





events and thus high intensity flows, i.e. floods (Yan et al., 2018). Reservoirs may require higher flood control due to the increased winter flows and increased risk to rain on snow events, while simultaneously requiring more storage capacity to counter decreasing summer flows. As summer low flows become more severe and snow-drought related summer streamflow droughts become more frequent, summer water scarcity may increase. The most severe water scarcity will likely occur due

to the coincidence of warm and dry conditions (AghaKouchak et al., 2014) and layered impacts from different drought types (Van Loon et al., 2015).

The GW-SW models in this study are generic, and, therefore, represent interpretive tools (Anderson et al., 2015). Like all GW-SW models, they are simplified numerical representations of natural flow systems and cannot duplicate the natural flow system exactly. However, the models are physically based, and the simulated streamflow is similar to observed

monthly streamflow in downstream watersheds. The consistency with previous studies indicates that the results can provide general insights into future water management challenges. Additionally, this study could be used as a base for identifying areas of interest and designing subsequent snow drought and streamflow drought modelling studies. While the models should not be used to forecast future water availability, results are discussed, in general terms, in relation to regional water management challenges in the following paragraphs.

In NEBC, where the Fort Nelson and Blueberry catchments are located, shifting snow and streamflow drought regimes will likely lead to decreased freshwater security. Since 2005, oil and gas industry development in NEBC has expanded rapidly due to advancements in hydraulic fracturing (Rivard et al., 2014). Hydraulic fracturing operations have short-term requirements for large quantities of water (Rivard et al., 2014), which puts high water demands on local watersheds. Without significant commitment on the part of industry to re-use and recycle water for hydraulic fracturing,

industrial water demand is likely to increase substantially – with the possibility of a more than 350% increase by 2030 compared to 2015 levels under a high development scenario (Kniewasser and Horne, 2015). Industrial freshwater abstractions are suspended during streamflow drought conditions, and the British Columbia Oil and Gas Commission issued water use suspensions six times between 2010 and 2019 (https://www.bcogc.ca/directives). As summer low flows decrease, water use suspensions are likely to become more frequent, and balancing increasing demand with decreasing security will be

a significant challenge for the region in the future.

In the Okanagan Valley, where the Whiteman catchment is located, surface water sources supply 67% of the annual water demand (Summit Environmental Consultants Inc., 2010). Most streams in the Okanagan are fully allocated, with no leeway for further allocations (Cohen and Kulkarni, 2001). The greatest proportion of water is used for agriculture, and irrigation, which accounts for 75% of regional consumptive water use, is expected to increase considerably with continued

climate warming (Neilsen et al., 2006). Additionally, average per person water use is high (Summit Environmental Consultants Inc., 2010) and population is expected to grow at a rate of 0.2 to 0.7% per year (BCstats, 2017). Population growth in the Metro Vancouver region, where the Capilano catchment is located, is expected to be even higher at 0.9 to 1.4% per year (BCstats, 2017). Significant opportunities exist for demand-side reductions in water use for the Okanagan (DHI Water and Environment, 2010; Neale et al., 2007) and Metro Vancouver (Metro Vancouver, 2011) regions. Water shortages



have already occurred both regions (Okanagan Water Stewardship Council, 2008; Metro Vancouver, 2015), and, considering the results presented in this study and others (DHI Water and Environment, 2010; Harma et al., 2012), summer water shortages are likely to be more common in the future.

## 6 Conclusion

5      Climate change impacts on snow drought, low flows, and summer streamflow drought were investigated using generic coupled GW-SW models for four headwater catchments in British Columbia. Results show that increased precipitation and temperature lead to decreased risk of dry snow droughts and increased risk of warm, and warm and dry, snow droughts. Climate warming and shifting snow drought regimes result in decreased summer runoff, decreased summer groundwater storage, and more severe summer low flows. Snow droughts have direct impacts on summer low flows, and temperature-related snow droughts not only become more frequent and severe in the future but are also more likely to result in summer streamflow drought conditions.

      The response of snow hydrology to climate warming is non-linear, and catchments with winter temperatures near 0°C exhibit substantially larger impacts from +2°C of warming compared to catchments with winter temperatures far below 0°C. The shift toward more frequent and more severe temperature-related snow droughts will decrease water availability during the summer for agricultural and industrial uses – potentially leading to decreased freshwater supply security, and the increased frequency of warm snow droughts will likely lead to an increased frequency of mid-winter melt events that will affect reservoir management. Changes in the low flow regimes will affect the ecology of river systems, and increased rain on snow events may require higher flood control.

**Data availability**

All data used is freely available and cited in the text.

The 10-day interval leaf area index (LAI) from Gonsamo and Chen (2014) was downloaded from: https://goo.gl/CTqaN7.

Daily climate time series for the climate change scenarios can be downloaded from PCIC's data portal, using the latitude and longitude of the catchments provided in Table 1.

**Author contribution**

JRD designed the study, built the hydrological models, and completed the analyses. JRD prepared the manuscript. JRD, DMA, and PW edited the manuscript.

**Competing interests**

The authors declare that they have no conflict of interest.





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
