# Peer review of "Climate change impacts on snow and streamflow drought regimes in four ecoregions of British Columbia"

_Hydrology and Earth System Sciences, 2019_

## Referee Comment (RC1) · Anonymous Referee #1 · 20 Mar 2020

General comments This manuscript aims to analyze the impacts of climate change on the evolution of snow and streamflow drought as well as their propagation relationship in four very small basins (<10 km2) in British Columbia. They used a combination of climate and hydrological models. However, I don't quite agree that there's enough innovation (similar to the mix of their two previous papers which have been published in WRR, such as Dierauer et al., 2018 and 2019) in this paper and even some methods may be problematic. Thus, the contribution of the current version of this manuscript is rather marginal and the study does not justify a novelty appropriately for HESS publication. Major concern: Although they describe the possible changes in drought propagation (from snow drought to streamflow drought), the full manuscript mainly focuses

on the individual changes of snow and streamflow droughts (e.g., the frequency and severity, very conventional drought characteristics). Actually, the drought propagation has its own property, such as the propagation time. What happened to the propagation time from snow to streamflow droughts and what might change in the future for the study regions. How to build the relationship (linear or non-linear) between snow drought and streamflow drought and analyze its changes? Unfortunately, I did not see anything like this. In addition, the authors did not consider the drought persistence (e.g., duration) when defining the snow and streamflow droughts. There may be cross-seasonal droughts. As a result, at this stage I am suggesting to reject (or major revise) the manuscript. Other comments: (1) Paragraph 10: the definition of snow drought looks more like meteorological drought (e.g., below-normal precipitation). (2) Table caption: the catchment characteristics should be more detailed, such as "Catchment characteristics (e.g., name, location, area, and etc.), including……". (3) The study basin area is too small, how accurate the downscaled is, especially for the daily climate data. Because the author uses the daily simulated streamflow series, the author needs to add the comparison results of daily downscaled and observed data (e.g., the precipitation and temperature). (4) Many previous studies have shown that the same climate models differ in streamflow simulations on different hydrological models. How the author considers it? The author should be adding more discussions. (5) Paragraph 25: hydrological drought » summer streamflow drought or streamflow drought? (6) The author should add the calibration and validation results of historical period (e.g., 1980s period). (7) 3.1.4 Section: Are the averages of climate model simulation results used and then analyzing drought results? Or each climate pattern with a drought result and then an average? Confused. (8) 3.2.1 Section: Confused. I don't know why this defines snow drought? For example, what is the below-normal peak snow water equivalent? Why use the term 'Years'? Since the differences in the catchment and climate features of four study basins are very obvious, why use the same $T25<0$°C (or $T25\geq0$°C) to define the "winter …" (or "summer …")? (9) Authors should pay attention to the difference between "streamfow" and "runoff". The full manuscript should be used in a

uniform manner. (10) 3.2.2 Section: Low flow is not equal to streamflow drought. How to consider the 'branch drought' in a long-lasting drought?

---

## Referee Comment (RC2) · Anonymous Referee #2 · 20 Mar 2020

Dierauer et al present an interesting study that aim to evaluate the impact of climate change on snow and streamflow drought in four small watersheds of British Columbia, partially based on the methodology presented in two previous studies published in WRR (2018 and 2019) by the same authors. To do so they set up a hydrological model MIKE SHE forced by statistically downscaled climate data under the RCP 4.5 and 8.5 scenarios and three GCMs. The mayor concerns I have about this paper, is how the model simulates the snowpack and the intrinsic assumptions and uncertainties that arise from this simplified approach (degree day). Realistically representing the snowpack is critical to the analysis the authors perform, as it is all based on the identification of snow droughts. I have serious doubt about the capacity of this model to properly

capture changes to the snow dynamic under the future scenario, or even if it can properly simulate historical snow (not shown). The authors argue that they only care about change to relative (historical vs projections) and not absolute changes (lines 13-15, page 9), then Figure 2 shows how not even relative changes are being well captured, not at least when compared to a more reliable energy balance. Note that this is not only about getting peak SWE right, but the timing is also not properly captures either (fig 2a vs 2b; differences between peak swe and depletion timing), as this is absolutely critical for groundwater recharge and streamflow, and the following overall analysis. I think the authors need to make a more compelling case, in which snow accumulation and melt (volumes and timing), and streamflow are being adequately represented by the model. Otherwise, any hydrological projections will lack scientific support. Therefore, following HESS's high publication standards, I recommend major revisions and resubmit when these problems are fixed. Below you will find more general comments that may help in your reviewed version of the manuscript: 1. A much-improved site description is needed. You emphasize groundwater however; no description of soils and geology is given. It is not clear why you chose these 4 small watersheds. How representative are these of each ecoregions, or they just happened to be in those ecoregions? If you are going to focus on the ecoregions more than the watersheds you need to describe them better, how do they differentiate and what's unique about them. Table1: values of climate are given yet there is no source for this information. Are there weather stations, stream gauge, snow pillow in these watersheds to compare the model with? 2. You emphasize that you are creating a "generic" model as an interpretative tool (line 23, page 5). I understand that, but even when doing so you need to show that the model can represent in a realistic way the hydrology. Even in a processes or physically based models, one can get very strange behaviors when using default values (as used in this study), you need to show this is not the case. 3. Many modeling decisions that need to be better supported, particularly in terms of parameters. 4. Can you include any streamflow and SWE observation to support your modeling decisions? 5. Figure 2: Include the other 2 watersheds. As is only the Blueberry site looks reasonable. Capilano

is very different between DD and EB. 6. Your definition of "winter" (page 10) is not very intuitive or clear, when do you starts counting to define your percentiles, January 1? Can you show some example of what does this translate into?

———————————————

---

## Referee Comment (RC3) · Matthew Cooper (Referee) · 25 Mar 2020

General comments: Dierauer et al. 2020 present an analysis of snow and streamflow drought in four catchments in British Columbia, Canada. They use an uncalibrated process-based hydrologic model forced with downscaled climate data from four general circulation models archived in the CMIP-5 climate model database. The process-based hydrologic model includes physical representations of unsaturated zone and saturated zone groundwater flow, physical representations of overland flow/channel routing, and conceptual representations of snow accumulation and snowmelt. This allows the authors to draw inferences about possible linkages between changes in snow

accumulation and melt, groundwater storage, and summer and winter low streamflow, including the lowest 20% of all summer and winter flows, referred to as "streamflow drought".

The authors justify their study by identifying two knowledge gaps: 1) no studies have explicitly quantified the impact of different snow drought types, i.e. dry, warm, or warm and dry, on the severity of summer/winter streamflow droughts, and 2) no studies have completed a combined analysis of snow drought and streamflow drought regimes in the context of climate change. Thereafter, the authors offer two hypotheses: 1) the snow-streamflow drought response depends on baseline mean winter temperature, and 2) the snow-streamflow drought response depends on changes in groundwater recharge/storage. Only the former is explicitly stated in the introduction, the latter is implicitly stated by the study design, reported results, and discussion.

The authors use four catchments that span a range of climatic conditions, from -0.6–5.9 oC average annual temperature, and 46–235 cm average annual total precipitation. This is appropriate with respect to the first hypothesis. The catchments also differ in terms of consumptive water use demands, which ends up dominating much of the discussion despite not being related to the study hypotheses. The catchment slopes range from 2–35o. Unsaturated zone and saturated zone depths, bulk soil densities, and hydraulic conductivity values are assigned based on published data sources. The combination of physically-based groundwater modeling, realistic soil properties, and variations in catchment slope are appropriate with respect to the second hypothesis.

The authors find that the distribution of future "snow droughts" into warm, dry, and warm and dry differ depending on the baseline mean annual air temperature and the future changes in air temperature and precipitation. The distribution of future low flows shifts toward less severe in winter and more severe in summer. Precipitation and temperature both increase under all future scenarios. The frequency and severity of dry and warm/dry snow droughts therefore decreases while warm snow droughts increase. Total annual runoff increases for all catchments because winter runoff increases more

than summer runoff decreases. Groundwater storage increases in winter/spring and decreases in summer. Actual evapotranspiration generally follows the same pattern.

Overall, the manuscript is well written, the methods are generally appropriate for the aims of the study, and the results are presented accurately. However, there are methodological issues, in some cases the results are poorly organized and/or unclear, and the link between the knowledge gap, the stated hypotheses, the results presented, and the discussion is either missing or is difficult to identify. For example, the stated purpose of the manuscript is to determine if future snow drought and streamflow drought depend on baseline climate conditions, yet the figure that most directly answers this question is given as the last figure (S9) in the Supplemental Information (an analogous figure for streamflow drought should be given). The literature cited is also a major weakness of the study, and the authors do not discuss their results in the context of prior work that has addressed the topic. Consequently, it is difficult to identify the most important contributions of this study in the context of prior work. The paper has potential to be a valuable contribution but will require major revisions. Below I provide specific major comments followed by a list of minor technical comments.

Specific Comments:

It is not clear why the changes in snowpack simulated by the hydrologic model MIKE-SHE are not used for the snow drought analysis and how those modeled changes compare with the GCM results presented in Section 4.1. This is critically important because the changes in modeled streamflow should depend strongly on the changes in modeled snowpack.

The authors justify the use of uncalibrated models by pointing to the risk of parameter overfitting and by arguing that relative changes are examined as opposed to absolute predictions. This is a controversial topic. In general, I agree with this justification but only if a detailed presentation of model performance is given. Without this it is impossible to determine if any credence should be given to the predictions of future snowpack

or streamflow conditions on which the study is based. The authors need to show comparisons between observed snowpack and streamflow and modeled snowpack and streamflow with MIKE-SHE.

The authors conclude that earlier snowmelt and higher winter snowmelt is linked to lower summer streamflow. However, Figure S2 shows that precipitation is higher in winter and in some cases is lower in summer, suggesting precipitation changes are also important. Similarly, the authors attribute changes in the seasonal distribution of groundwater recharge/storage to changes in snowpack without disentangling the effect of precipitation changes. The authors need to conclusively demonstrate the connection between changes in snowpack accumulation/melt and changes in summer streamflow. The authors should be able to attribute exactly how much snowpack changes contribute to streamflow changes and similarly for AET and PPT. This is the main conclusion of the paper and therefore it must be absolutely clear to the reader how this conclusion is drawn.

Along these same lines, too much emphasis is given on the link between snow drought and groundwater-surface water dynamics in the abstract, introduction, and methods, given the lack of emphasis on this topic in the results and discussion. Owing in part to the missing water balance attribution noted above, this study does not give a process explanation or evidence of direct linkage between changes in snowpack and changes in groundwater and/or low flows. A key assertion of this study is that it demonstrates the 'direct impact' of snow drought on summer low flows, but those direct impacts were not demonstrated in this study.

There is also a large and rapidly expanding body of work that is not discussed. In some cases, the cited literature is self-referential where other studies could also be included. A more detailed review of prior work might help the authors address the previous comments about groundwater-streamflow-snowpack linkages, and a list of works that should be referenced and appropriately discussed is included in the comments below.

Technical comments:

P1 L16 – what is meant by "generic"? Perhaps replace with a specific term such as "physically-based" or "process-based"?

P1 L21 – snow droughts propagate into summer streamflow droughts more frequently in the 2050s and 2080s as compared to the baseline 1980s period. This is an important result. Take a look at the study by Marshall et al. 2019 Geoph. Res. Lett. It might be appropriate here.

P1 L27: "more likely to fall as" is vague and unphysical, replace with "the rainfall fraction of precipitation is expected to increase relative to snowfall" or similar. Also please include a citation – suggest Gergel et al. 2017 Clim. Chang but there may be a better choice.

P2 L4: Throughout the introduction but especially here (following " ... during the summer low flow period"), the large body of work that has examined interactions between snowmelt, groundwater, and summer streamflow need to be referenced. Key missing papers include:

Tague et al. 2008 Clim. Ch. Jefferson et al. 2008 Hyd. Proc. Tague and Grant, 2009 WRR Safeeq et al. 2013 Hyd. Proc. Tague and Peng, 2013 JGR Safeeq et al. 2014 HESS Kormos et al. 2016 WRR Jepsen et al. 2016 JoH Cooper et al. 2018 WRR Cummings and Eibert 2018 J. Water. Resou. Son and Tague, 2018 Ecohydrology as it relates to AET/SWE Possibly various works of Reed Maxwell and his students

P2 L8: Throughout this paragraph, several papers that examine snow drought, including the link between snow drought and climate change, are not referenced:

Cooper et al. 2016 ERL Sproles et al. 2017 HESS Hatchett and McEvoy 2018 Earth. Int. 2018 Marshall et al. 2019 GRL The authors might also alert the readership to the difference between hydrologic drought and meteorological drought, and point to drought indices that include snow e.g. Staudinger et al. 2014 WRR.

P2 L17: "will likely lead to" Please add at least one citation to support this statement.

P2 L19: "its baseline mean winter temperature". Please cite Nolin and Daly 2006 and Tennent et al. 2015 and confirm if others should be cited as well. Here (as elsewhere) there is only one reference to prior work, and it is self-referential.

P2 L20: "generic" please see prior comment

P2 L30: It might help the reader if the descriptions are consistent i.e. the climate, ecoregion, and surface water supply, whereas as now snow (depth? SWE?) is noted as <15 cm for the first two but not for the second two. Also, "relatively cold, dry winters" is unclear – just state the mean annual T and P, use the same sentence structure so the reader can quickly mentally categorize the four catchments.

P2 L32: use consistent precision, assume this means 0.0 oC

P5 L21: would it be more appropriate to say, "The GW-SW model scenarios in this study were developed ..."? At present, it sounds like the actual models were developed, but instead the models were used to develop modeled states for a range of climate forcing i.e. scenarios. This might just be a semantic issue, but I think most readers will interpret 'model development' to mean that a new model was written.

P5 L24: Ahh ... now I understand "generic". I think "uncalibrated" is better unless "generic" is accepted terminology.

P6 L14: "therefore, Manning's n values from" should read "Manning's M is the reciprocal of Manning's n and was estimated from values in Chow (1959) (Table S1).

P6 L17 "is modelled"

P6 L22: delete "zones" after UZ and SZ

P8 L7: "within the range"? perhaps choose something more specific, like the median? I agree it shouldn't matter much but it seems odd to pick a random value.

P8 L20: How is snowfall simulated?

P8 L22 "The degree- day method does not, however, account for several factors that are important for snowmelt, including wind speed, humidity, topography (slope, aspect, and shading), cloud cover, and vegetation" It is more important that the degree-day method does not account for net solar radiation and net longwave radiation. These other factors listed are important but less so. It is also not convincing to cite a WMO paper from 1986 to justify the performance of degree day models relative to energy balance models. The authors need to find a robust set of references that support this statement, which should acknowledge the strengths and weaknesses of the degree-day method.

The comparison between MIKE-SHE and CHRM is important and I thank the authors for performing this comparison, but perhaps the details can be moved to the SI and replaced with a shorter justification. Also, it is unfortunate that two of the four study catchments were left out. It might be justified to leave out Fort Nelson given its proximity to Blueberry, but Whiteman has a unique geographical/climatic setting.

Figure 2: This figure is difficult to interpret. Please replace "EB" with Energy Balance and/or CHRM, replace DD with degree-day, reorganize the figure so that DD and EB are on the same plot i.e. six figure panes are ok, but the first should be DD vs EB during baseline, the second should be DD vs EB in 2050s, and the third should be DD vs EB during 2018s. The purpose is to compare the models.

P9 L12: "The emphasis of this study, however, ..." on the other hand, this study also focuses on links between snow, streamflow, and groundwater recharge/storage, and the latter very much depends on land cover type. Therefore, this justification somewhat undermines the methods. Instead of this justification, I think it is best to just state the strengths and weaknesses, so the reader is fully aware.

P11 L13: the issue of summer rainfall might need to be given more treatment to convince the readers that changes in snowpack are driving the changes in low flows

P12 L5: this material should be moved to the methods section

P13 L2: Looking at Figure S5, it appears that peak SWE decreased for Fort Nelson. Are the box plots in Figure 3 based on the sum of all snow mass in the catchment, whereas the curves in figure S5 are average SWE depth? If so, you should make them consistent. The box plots show an increase in SWE, whereas that is not evident in figure S5.

Section 4.1 P12 L25 – P13 L25 I got confused in this section because I expected model results from MIKE-SHE. Not sure if you need to change it, but just consider that this could confuse your readers. I am not sure how much the GCM water balance really matters, since the streamflow results are from MIKE-SHE.

P13 L15: discussion of AET changes in summer should reference Bales et al. 2018 Sci. Rep., Kormos et al. 2016 WRR, Cooper et al 2018 WRR, Goulden and Bales 2014 PNAS and others esp. the work of Tague.

P13 L23: confirm that the CMIP5 models examined are not those with a dynamic carbon cycle (I think those were not included in CMIP5 but this should be confirmed as per the statement that the results do not reflect changes in vegetation)

P14 L4: Please also cite Tague and students that find same results, including figures that show the same behavior as your figure S7 and S8 (e.g. Fig 11 Tague et al. 2008 Clim. Chang). The review of Klos et al. 2017 Water might also be a good reference.

Table S6 does not show the ratio R:P that is referenced in the text.

Figure 4: This result that dry snow droughts disappear and become warm snow droughts is very interesting! It also shows that for the coldest catchment there is no change in severity or frequency, also very interesting, great work. I think this could be emphasized a bit more, perhaps in the abstract, something along the lines of "our results suggest that even under an extreme climate warming scenario, snow drought risk may not increase in catchments below certain thresholds of winter temperature".

P17 L6: "Snow drought risk" has not been defined yet. Perhaps add a parenthetical definition as in the table?

P17 L10: the non-linearity of snow sensitivity has been shown in other studies including Luce et al. 2014 WRR Fig. 4 and Fig. 5 among others.

P18 L9: is it true that "anomalously low flows are equivalent to streamflow droughts"? A citation should be included in either case.

Section 4.3: Throughout this section, as in the introduction, there are many prior studies that should be acknowledged, in addition to the Dierauer et al. 2018 study. Studies that explicitly examined connections between snowpack and summer low flows include Jenicek et al. 2018 WRR (in addition to the Jenicek 2016 paper that was cited in the introduction), Cooper et al. 2018 WRR, Kormos et al. 2016 WRR, and Jepsen et al. 2016 JoH

P21 L9: replace "generic" with "uncalibrated" or similar

P21 L13: If you are focusing on the GW-SW changes, then a compact version of Figure S7 or S8 needs to be moved to the main. Perhaps just show Capalino compared with one of the others to show the contrasting behavior, and just show one scenario.

P22 L2: This is another opportunity to reference prior work.

P22 L20 – P23 This is a very good discussion of impacts.

P23 L10: this comes as a bit of a surprise this far into the manuscript. The modeled streamflow should be compared to the observed streamflow on the same timescale that the analysis is reported, and a figure should be included in the results section. As elsewhere, it's ok if the model isn't perfect, but the readers should know the strengths and weaknesses of the model relative to observations.

P23 L11: I am not sure about the claim that this study is useful as a base for identifying future studies. This study reports on four very small catchments in Canada. Instead,

you might highlight the model approach, and the importance of using GW-SW models when examining streamflow response to snowpack changes. Along those lines, it seems like you have a real opportunity to leverage the detailed UZ/SZ groundwater modeling included in MIKE-SHE to carefully disentangle how changes in snowpack propagate into changes in summer low flows. If you quantify the connection between changes in snowpack and changes in low flows, the general conclusions would be supported more convincingly. You might point to prior regional-scale empirical studies such as Safeeq et al. 2013 and then highlight that your study provides a template for the design of future modeling studies that put those empirical and theoretical observations to the test.

P24 L16: Is it accurate to say that more frequent warm snow droughts will lead to more frequent mid-winter melt events? Or are they both caused by the same thing? Maybe rephrase as "increased frequency of warm snow droughts will likely be associated with an increased frequency of mid-winter melt events ...".

Thank you for the interesting study. Please feel free to contact me with any questions.

Matt Cooper guycooper@ucla.edu